# Investigation of interface characteristics and mechanical performances of Cu/Al plate fabricated by underwater explosive welding method

**Jinhua Chen[1,2], Dapeng Zhou[1,2]\*, Xiaohong Zhou[1,2], Hanliang Liang[1,2], Peizhong Feng[3], Yonghao Yu[3]\*, Xueqin Kang[3]\***

1 Huaibei Blasting Technology Research Institute Limited Company, China Coal Technology Engineering Group, Huaibei, China, 2 Anhui Key Laboratory of Explosive Energy Utilization and Control, Huaibei, China, 3 School of Materials Science and Physics, China University of Mining and Technology, Xuzhou, China

\* hbzhoudapeng@126.com (DZ); yuyrrow@163.com (YY); cumtkxq@cumt.edu.cn (XK)

## Abstract

Cu/Al composite plate was manufactured by underwater explosive welding method. The interface characteristics and mechanical properties of Cu/Al composite plate were evaluated and analyzed through phased array ultrasonic inspection, microstructure, uniaxial tensile test, three-point bending test, tensile shearing test and microhardness test. The results showed that the welding of thin Cu and Al plates is achieved by underwater explosive welding, with a Cu plate thickness of only 0.5 mm. A well bonded interface between Cu and Al plate is obtained, at a detonation velocity of 4 000 m/s, when the distance between Cu foil and Al plate is 0.2 mm. There are wavy fusion zones at the bonding interface of Cu/Al composite plate. No delamination or cracks are found at the bonding interface between Cu and Al during tensile and bending tests, and local cracking only occurs at the necking part in the tensile test due to severe deformation. The tensile strength and minimum tensile shearing strength of Cu/Al composite plate reaches 133 and 72.9 MPa, respectively. The hardness values of fusion zone, Cu and Al at the interface reach 385, 135 and 52 HV, respectively. The increase in hardness of Cu and Al near the interface is mainly caused by severed deformation induced by intense shock pressure.

## 1. Introduction

Copper (Cu) and Cu alloys have high strength and excellent conductivity, good thermal conductivity, and well oxidation resistance, consequently, they have been widely used in industries such as electrical, electronic, chemical, power and military [1,2]. Cu accounts for only 0.01 wt. % of the Earth's crust and is difficult to extract and refine. Cu plate and strip cannot meet the needs of China's economic development, and China still needs to import a large amount of Cu plate and strip from abroad every year [3,4]. There is an urgent need to find materials with excellent performance to replace Cu and Cu alloys, and this can increase corporate profits and alleviate dependence on Cu resources. The content of aluminum (Al)

**Data availability statement:** All relevant data are within the manuscript and its Supporting Information files.

**Funding:** This work was founded by the Key Research and Development Plan of Anhui Province (2022a05020021). The funders had no role in study design, data collection and analysis, decision to publish, or preparation of the manuscript.

**Competing interests:** The authors have declared that no competing interests exist.

in the Earth's crust is about 8 wt. %, making it the most abundant mental and the third most abundant element [5]. Al has a conductivity of $3.5 \times 10^7$ S/m and only lower than that of silver (Ag), Cu and gold (Au). The density of Al is $2.7 \, g/cm^3$, less than one-third of the density of Cu ($8.9 \, g/cm^3$) [6]. Al and Al alloys have excellent antioxidant properties in atmospheric environments due to the timely formation of a dense aluminum oxide ($Al_2O_3$) film with strong adhesion on their surface.

Cu/Al composite materials not only have the merits of high conductivity and thermal conductivity of Cu, but also have the merits of light weight, corrosion resistance, economic, and pleasing to the eye. They are mainly used in Cu-Al transition joints, Cu-clad-Al cables, Cu-Al heat dissipation fins, electronic packaging and other fields [7,8].

The main processing methods for Cu/Al composite include rolling composite [9], surface spraying [10], extrusion drawing [11] and vacuum brazing [12]. Rolling and extrusion drawing are only used to fabricate plates or rod materials, separately. Surface spraying belongs to thermal processing, with high temperature, and low melting point metal powder Al is prone to volatilization, making the working environment harsh. Vacuum brazing uses brazing materials with lower melting temperature, resulting in lower connection strength and more interfaces. Cu and Al are prone to form various intermetallic compounds, due to the low mutual solubility and significant differences in physical and chemical properties. The generated intermetallic compounds will reduce the performance of Cu/Al composite materials, form cracks and lead to fracture failure during service process [13].

Explosive welding is a metal joining technology in solid state. It uses high-speed oblique impact to produce welded joints with the help of controlled explosive forces induced by explosive charges [14]. Explosive welding joints have very strong adhesive strength compared with the joints manufactured by other technologies [15]. Explosive welding is mainly used for welding metals with significant differences in metallurgical properties. These metals, such as titanium (Ti)/steel [16], Al/steel [17], niobium (Nb)/steel [18], Cu/steel [19], molybdenum (Mo)/Cu [20], Cu/Al [21], magnesium (Mg)/Al [22], Ti/Al [23], are difficult to join through other approaches. However, traditional explosive welding technology is not suitable for welding very thin metals, due to the enormous explosive force and high temperature generated during the traditional explosive welding process, which can destroy the sheet metals. The thickness of the sheet metals used in the above researches generally exceeds 2 mm. Underwater explosive welding can be used for welding very thin plates with a thickness of 0.5 mm or even lower, because water acts as a transport medium to generate a uniform pressure at the position where it is needed. In addition, the incompressibility of water is much higher than that of air, because the density of water ($1000 \, kg/m^3$) is 800 times that of air ($1.29 \, kg/m^3$). These characteristics of water ensure that the explosive force does not generate high temperature in water, but it does enable the water to move forward together with the foil metals, and maintaining their integrity. The peak value of underwater shock waves is higher than the pressure impact of air waves, due to not consuming the impact energy generated by explosive and incompressibility of water. This allows the foil to achieve a high velocity in a short distance, and enough to generate the high energy required for metallic bonding [8,24]. So, underwater explosive welding can become a successful thin plate welding pattern.

As of right now, different foil metals such as Cu/NiTi [14,25], Cu/Al [26], Mg/Cu [27], amorphous metal/Cu [28], Ti/Cu [29], Ti/Mg [15], zinc (Zn)/Al [30], Al/steel [31] and steel/ stainless steel [32] have been produced successfully through underwater explosive welding. The thickness of the flyer plate is generally 0.5mm, and even reaches 0.03 mm. The thinning of Cu plate thickness saves more strategic resources Cu and reduces material costs. Some researches on underwater explosive welding only presented the macroscopic [25] or microscopic [15,28,30] morphology of composite material cross-sections without studying their

bonding interface and properties. Some researches have investigated the performances of composited materials, but only focused on the hardness at the bonding interface [26,27] and the tensile [27,29], bending [26,27] or compressive [31] properties of the composite material. Extensive researches on underwater explosive welding have focused on the distance of flyer plate to lower plate, flyer plate velocity, collision point speed, collision angle and the usage of explosives [14,15,25,28]. Furthermore, the morphology changes and welding changes of lower plate or flyer should be concentrated, especially the boding force between lower and flyer plate.

In present research, Cu/Al composite plate was prepared through underwater explosive welding method. After welding, the bonding quality between Cu foil and Al plate was inspected using phased array ultrasonic test for the first time. In addition, uniaxial tensile and three-point bending test were used to characterized the tensile and bending properties of composite plate. Tensile shearing test was first used to characterized the bonding strength between Cu foil and Al plate. The bonding interface characteristics were measured using a microhardness tester, the morphology and element distribution of the bonding interface were observed.

## 2. Materials and experiments

### 2.1 Base materials

Cu foil and Al plate sectioned in 200 mm × 500 mm × 0.5 mm for the flying and 200 mm × 500 mm × 5 mm for the base specimen. The chemical composition of Cu foil and Al plate was measured using emission spectrometry. Test results of Cu foil and Al plate are represented in Table 1, The compositions of Cu and Al were meet the requirements of T2 and 1060.

### 2.2 Underwater Explosive Welding Device and Process

Underwater explosive welding device, as shown in Fig 1 was employed as the test system. The mixed powder explosive with a detonation velocity of 4 000 m/s was used during the underwater explosive welding process. To ensure the efficiency of explosive welding, all devices were submerged in water. The Cu foil was shelved and separated from the Al plate using shims

Table 1. Chemical composition of Cu and Al (wt. %).

| Material | Cu | Fe | S | Zn | Si | Mn | Mg | Al |
|---|---|---|---|---|---|---|---|---|
| Cu | 99.989 | 0.004 | 0.007 | – | – | – | – | – |
| Al | 0.003 | 0.153 | – | 0.041 | 0.098 | 0.012 | 0.009 | 99.684 |

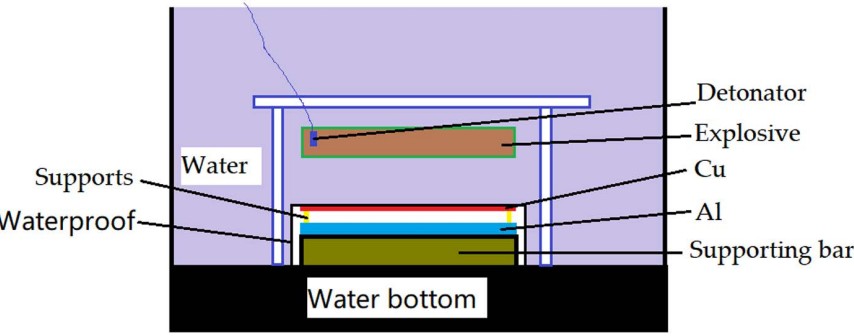

Fig 1. Schematic diagram of underwater explosive welding device.

stuck at the four corners. The air gap between the Cu foil and Al plate was 0.2 mm and sealed against the ingress of water. The supporting bar with Al plate, shims and Cu foil was put into a seal bag worked as waterproof. The materials of supporting bar and waterproof were steel and polyethylene, separately. Water bottom was made of sand and worked as an anvil and buffer to ensure the stability of the device. Thickness of water layer between the Cu and the explosive was 10 mm.

## 2.3  Phased array ultrasonic inspection

Phased array ultrasonic inspection is a non-destructive testing technique that utilizes the propagation characteristics of ultrasonic waves within materials to detect internal defects. The basic principle involves a straight probe emitting ultrasonic pulses perpendicular to the inspection surface of the workpiece. Ultrasonic waves reflect when they encounter interfaces between different media (such as cracks or pores). The reflected ultrasonic signals are received by the same probe. By analyzing the time, intensity, and other information of these echoes, it is possible to determine whether there are defects in the material and ascertain their location, size, and other characteristics [33,34]. This method is suitable for inspecting materials with uniform thickness and flat surfaces. In the phased array ultrasonic test, a phased array device ISONIC-USD3–5 manufactured by Sonotron NDT was used, with a probe model of 5 MHz and Φ10 mm. The basic parameters were set to a gain of 34.5 dB and a sound path of 2.5 mm. The probe was operated in dual-crystal mode for excitation, with a pulse height of 85 ns, impedance of 25 Ω, and a repetition frequency of 565 Hz. Gate A was positioned at 5.6 mm with a threshold height of 33%. The scanning parameters included a scan length of 160 mm and a scan speed of 30 mm/s, with an industrial slurry used as a coupling agent. Scans were conducted on the underwater explosive welding composite plate at the starting, middle, and ending regions to assess the bonding quality and defect distribution in different areas of the joint.

## 2.4  Microstructure characterization

Samples with dimensions of 10 mm × 10 mm × plate thickness were sliced from Cu, Al and Cu/Al joint plates using a wire-cut machine. Preparation of metallographic specimens mechanical grinded using 180#, 400#, 800#,1500# and 2000#-grit waterproof SiC sandpaper, and then the grinding surface was polished by the diamond paste, till the surface of the specimen was polished to a mirror with a roughness of 1 μm. Finally, the polished surface was thoroughly scoured by anhydrous ethanol and dried with a hairdryer. Cross section morphology of Cu/Al joint was examined and captured by an Olympus optical microscope (OM). Then, the polished specimen surfaces of Cu and Al was chemically etched using a mixture of Glyceregia (mixture of glycerol, nitric acid and hydrochloric) and hydrofluoric acid, respectively. The microstructures of Cu foil, Al plate and Cu/Al joint interfacial were also displayed by OM.

## 2.5  Uniaxial tensile test

Uniaxial tensile tests of Cu, Al and Cu/Al composite plates were conducted on an electronic universal test machine (UMT5305) at normal temperature under a stretch speed of 1 mm/min according to the standard ASTM E8/E8M-22. Equation (1) was used to calculate stress of Cu foil, Al and Cu/Al composite plates. As can be seen from Fig 2, dog-bone shaped tensile sample was cut from the Cu, Al and Cu/Al composite plates. Detailed parameters of tensile test sample were marked in Fig 2. After tensile test, fracture surface of plate was examined by scanning electron microscope (SEM, FEI Quanta TM 250). The sample chamber was

evacuated to low vacuum before testing. The voltage was adjusted to 15.0 kV during testing. A typical picture was selected for Cu, Al and Cu/Al materials to analyze.

$$\sigma = F / \left(15 \ mm \times b\right) \tag{1}$$

Where F is the applied load in [N], b is the thickness of Cu, Al or Cu/Al composite plates in [mm].

## 2.6  Three-point bending test

Three-point bending test of Cu/Al composite plate was carried out on an electronic universal test machine at room temperature according to the standard ASTM E290-22. To evaluated the connectivity between Cu foil and Al plate, the Cu/Al composite plate was bent into a U-shape from a 10 mm × 100 mm × 5.5 mm strip specimen at displacement rate of 5 mm/min perpendicular to the welding direction. Schematic diagrams before and after three-point bending test are shown in Fig 3.

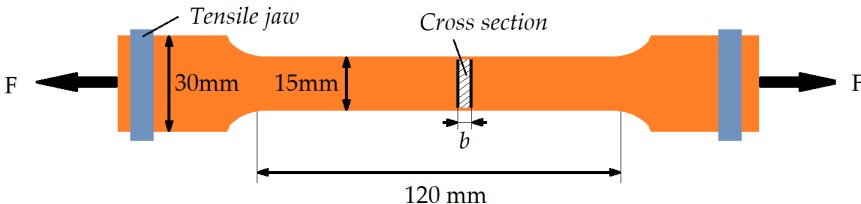

**Fig 2.  Schematic diagram illustrated uniaxial tensile sample and test of Cu/Al composite.**

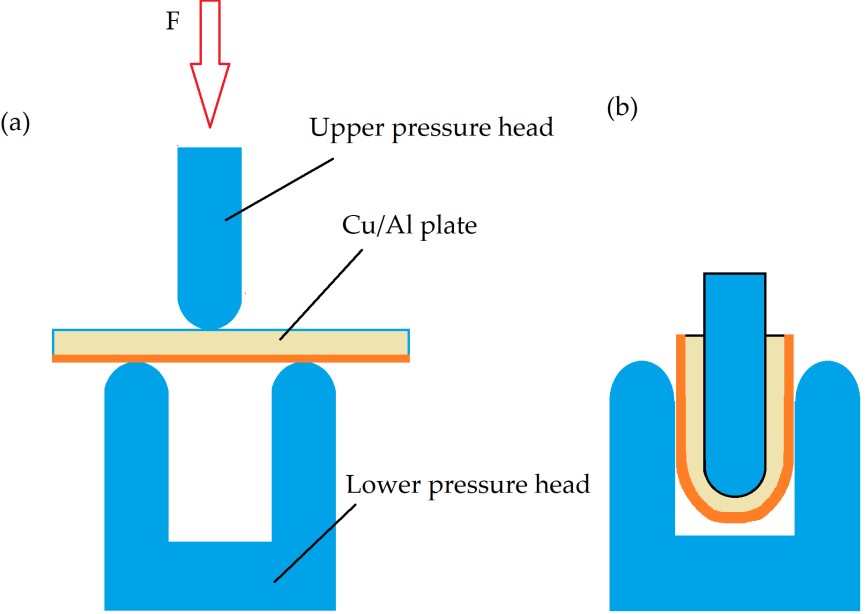

**Fig 3.  Schematic diagrams of Cu/Al composite (a) before and (b) after three-point bending test.**

### 2.7 Tensile shearing test

The tensile shearing test was conducted on an electronic universal test machine at room temperature to test the interface bonding strength of Cu/Al composite plate. The specimens were machined by wire cutting, and the schematic diagram is shown in Fig 4. The specimen dimension is 10 mm × 100 mm × 5.5 mm, and the notches wide 2 mm and 4 mm are machined on the Cu foil and Al plate, respectively. The overlapping area between Cu foil and Al plate is 1 mm × 10 mm. The two ends of the specimen are clamped and subjected to a tensile shearing test at a speed of 0.5 mm/min. The shear strength of Cu/Al composite plate is equal to the peak load during the tensile shearing test provided by the overlap area. This value can reflect the bonding performance of Cu foil and Al plate.

### 2.8 Microhardness analysis of Cu/Al composite plate

Microhardness measurements were conducted on the underwater explosive welding Cu/Al composite sample using the WILSON hardness tester (VH1102) according to the standard ASTM E92-23. The test load was set as 0.01 kgf, with a dwell time of 15 seconds. Hardness gradient data were obtained across the bonding interface to analyze the hardness variation of the Cu/Al composite interface. Test points were arranged at intervals along the interface region to provide a detailed characterization of the hardness gradient distribution.

## 3. Results and discussion

### 3.1 Analysis of phased array ultrasonic inspection

Fig 5 shows the morphology of the Cu/Al composite plate. The Al plate and Cu foil are tightly bonded together, and there are no cracks between them on the side. Phased array ultrasonic inspection was performed on the entire board. Fig 6 presents the typical results of the phased array ultrasonic inspection at starting (Position S), middle (position M) and ending (position E) location of Cu/Al composite plate. The width and length of the scanned composite

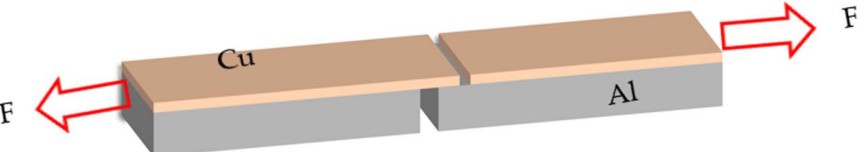

**Fig 4. Schematic diagram illustrated tensile shearing specimen and test of Cu/Al composite.**

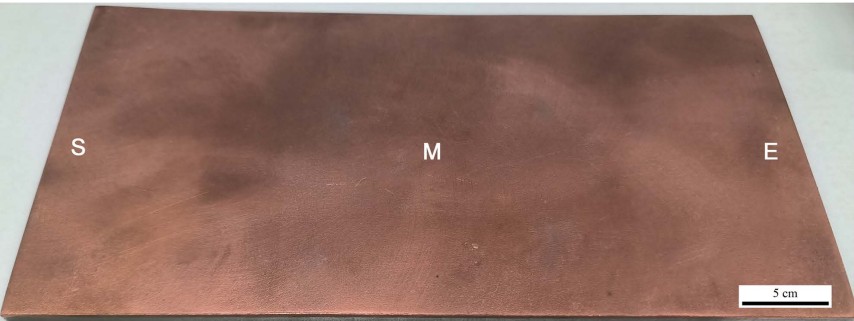

**Fig 5. Morphology of Cu/Al composite plate and detect position of phased array ultrasonic inspection.**

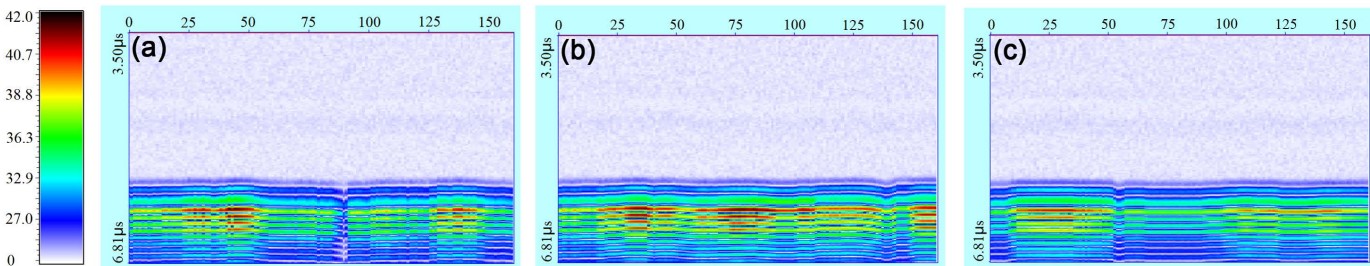

**Fig 6. Phased array ultrasonic inspection images of the (a) starting, (b) middle and (c) ending point of underwater explosive welding Cu/Al composite plate.**

materials are 5.6 mm and 500 mm, respectively. The width dimension is equivalent to the diameter dimension of the probe. In the images, the upper light-colored regions represent the Cu/Al composite plate manufactured by underwater explosive welding. Clear image images can be observed in all positions [35]. No significant abnormal echo signals were observed during the inspection, indicating a well-bonded interface with no detectable cracks, porosity, or delamination [36–39]. The ultrasonic signal intensity remained stable across the interface region, with consistent echo reflection patterns, suggesting a uniform bond quality achieved by the underwater explosive welding process. These results meet the design requirements for weld quality.

## 3.2 Microstructure and welding interface morphology

Fig 7 shows the metallographic structures of Cu and Al samples under an optical microstructure. As shown, the microstructures of Cu and Al are equiaxed geometry before underwater explosive welding, and there is a small amount of twinning in the Cu samples.

Fig 8 shows the interface appearance formed by underwater explosive welding. The bonding interface between Cu and Al was formed by some disordered waves. A localized fusion area was formed at the interface between Cu and Al during the welding process [40]. During the underwater explosive welding process, Cu foil collides with Al plate at high speed, converting kinetic energy into thermal energy, and resulting high temperature and pressure caused local melting of Cu and Al, and subsequent rapid cooling to room temperature instantly during underwater explosive welding. Lin et al. [31] and Satyanarayan et al. [30] reported the

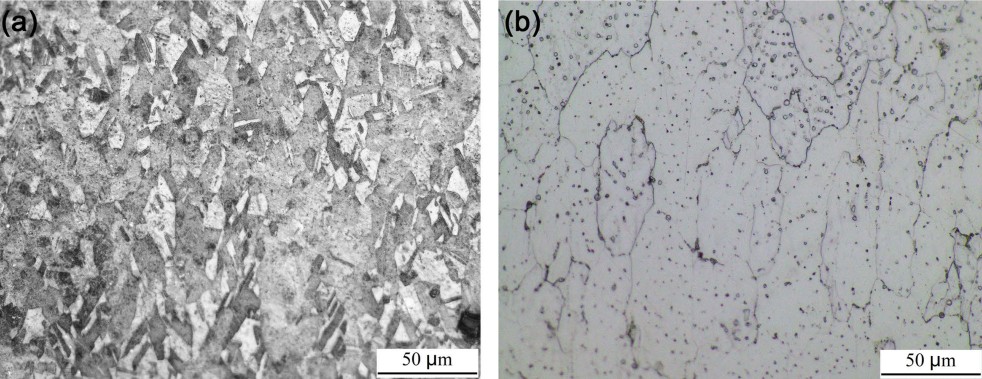

**Fig 7. Microstructure of (a) Cu foil and (b) Al plate for underwater explosive welding.**

same fusion area between Al/Steel and Sn/Al composite materials fabricated by underwater explosive welding. Fig 9 shows the morphology and energy dispersive spectrometer (EDS) of the fusion zone at the interface between Cu and Al. In addition to the wavy Al deformation zone at the interface, there were also a certain fusion zone formed between Cu and Al. Table 2 shows the EDS analysis of the points marked in Fig 9. The test results revealed that only Cu and Al elements were present in Cu foil and Al plate, respectively. The atomic ratio of Cu and Al elements in point B was 34.42 to 63.58, indicating the formation of the intermetallic

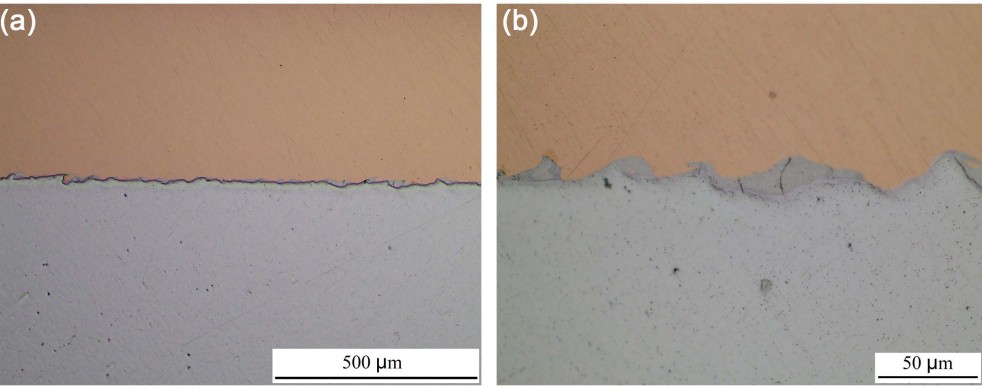

**Fig 8. (a) Low and (b) high magnification morphology of Cu/Al composite interface for underwater explosive welding.**

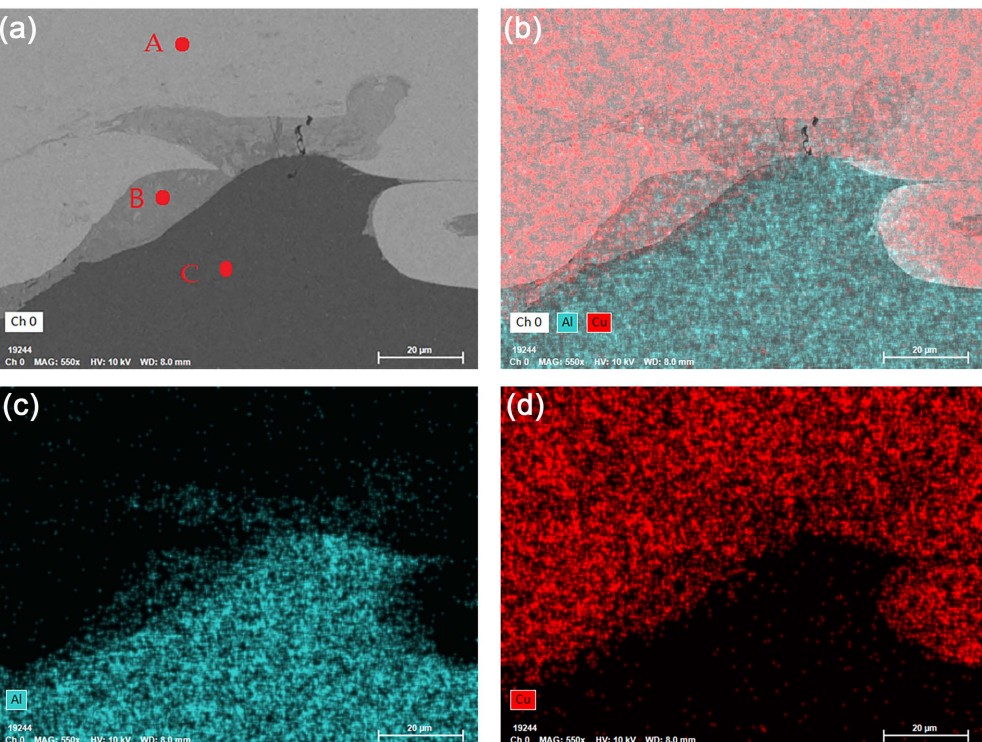

**Fig 9. (a) SEM image, (b) EDS, and (c) Al, (d) Cu element distribution of Cu/Al composite interface for underwater explosive welding.**

**Table 2.  The EDS analysis of different points presented in Fig 8.**

| Position | Al (at. %) | Cu (at. %) | Cu-Al phase |
|---|---|---|---|
| Point A | – | 100% | Cu |
| Point B | 63.58% | 34.42% | $CuAl_2$ |
| Point C | 100% | – | Al |

compound $CuAl_2$. Gladkovsky et al. [41] and Honarpisheh et al. [42] reported the intermetallic compounds formed in Cu/Steel and Al/Cu/Al composites produced by explosive welding.

Fig 10 shows the metallographic structures of Cu foil and Al plate around the joint interface under an optical microstructure. As shown, some grains undergo plastic deformation (only a few micrometers) due to the explosion waves. The microstructures of Cu and Al located slightly away from the bonding interface were the same as that of the material before underwater explosive welding.

### 3.3  Mechanical characterization – tensile test

Fig 11 displays the stress-strain profiles of Cu, Al and Cu/Al composite plates. The tensile strengths of Cu, Al and Cu/Al composite plates were 268, 98 and 133 MPa, respectively. The curves in Fig 11 are divided into three regions coordinating with different stages of Cu, Al and Cu/Al composite plates tensile test. The first stage is elastic stage. The length of the tensile specimen grows with the stress growing, but the sample restores to is primitive size when the stress fully unloaded. The second stage is uniform plastic deformation stage. The tensile specimen continues to elongate and the stress remains constant or reduces slightly. The deformation leads to a decrease in the cross-sectional area of the tensile specimen and material work hardening, resulting in an increase in the length of the tensile sample, and a maintenance or

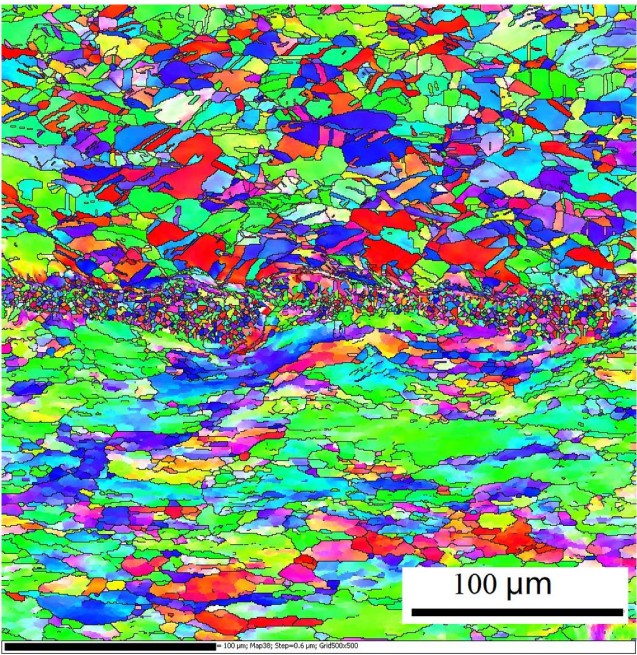

**Fig 10.  Microstructure of Cu foil and Al plate near the bonding interface of Cu/Al composite for underwater explosive welding.**

slight decrease of stress. The third stage is concentrated deformation and fracture stage. The stress rapidly decreases, and eventually the tensile specimen breaks, with the stress sharply dropping to zero, due to the occurrence of concentrated deformation. Fig 12 shows the tensile fracture surface SEM morphologies of Cu and Al samples. There are significant differences in the thickness of Cu, Al and Cu/Al samples before and after tensile test due to massive

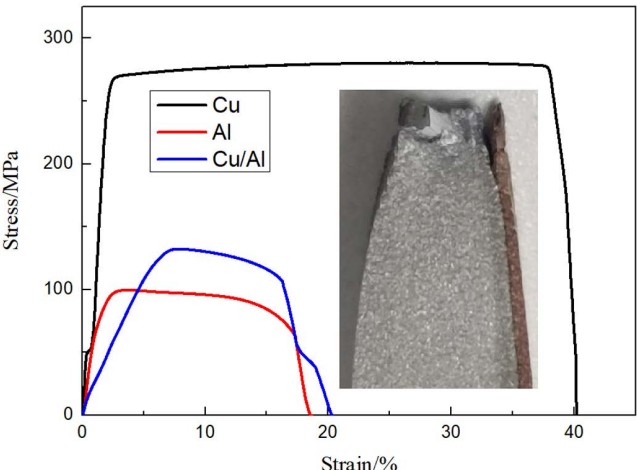

**Fig 11. Stress-strain curves of Cu, Al and Cu/Al composite plates.**

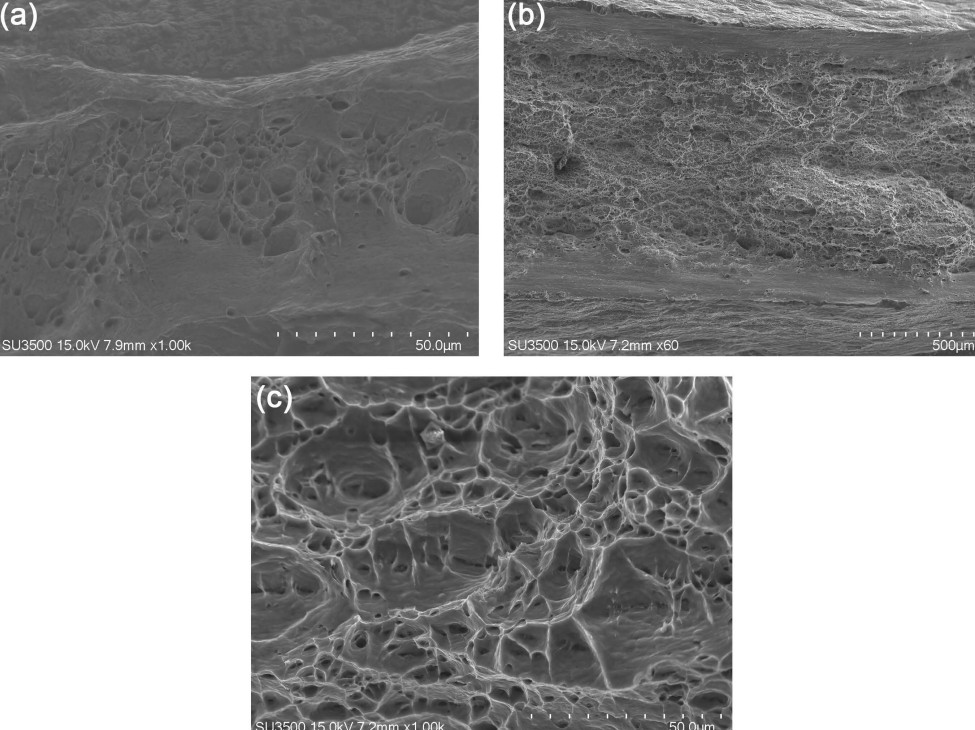

**Fig 12. The entire thickness fracture SEM morphologies of (a) Cu and (b) Al tensile samples and (c) high magnification of Al fracture surface.**

deformation. The microstructure of the Cu and Al fracture surfaces show equiaxed dimples, indicating that both Cu and Al have good plasticity, which is consistent with the significant deformation generated during the tensile test. Cu/Al composite plate exhibits a curve that matches that of Al plate, due to the thickness of Al plate. From the tensile curves of Cu foil and Al plate, it can be seen both have good plasticity, but the elongation of Cu is higher than that of Al. Cu/Al composite plate exhibits cracking at the bonding interface due to their different deformation abilities caused by significant plastic deformation in the concentrated plastic deformation area. In other area bonding interface, there is uniform plastic deformation and no cracking occurs. The fracture surface of Cu/Al composite plate is uneven, exhibiting the same characteristics as the Cu foil and Al plate fracture surfaces in Fig 12. Sun et al. [27] reported the less delamination in the tensile fracture of Al/Cu composite plate fabricated by underwater explosive welding. The tensile fracture surfaces of Cu and Al exhibited typical dimple characteristic.

## 3.4  Bending deformation characterization – Three-point bend test

The three-point bending test on the underwater explosive welding Cu/Al composite material was conducted and shown in Fig 13. The images were captured every 2 min. There was no separation or cracking at the interface between Cu and Al interface during the bending procedure, even under maximum bending deflection. Fig 14 shows the whole and local appearance of Cu/Al composite plate after bending test. The entire specimen maintained structural integrity. There were no delamination or cracking defects, even in the area with the largest deformation. These results confirm that there is an excellent bonding interface between Cu and Al in underwater explosive welding, and the bonding between Cu and Al will not be destroyed under large deformation. The absence of delamination and cracks at the bonding interface during and after bending test indicates a strong metallurgical bond between Cu and Al plate, which is crucial for application of Cu/Al composite materials in environments requiring high load-bearing capacity. The integrity exhibited by Cu/Al composite plate in the bending test demonstrates the reliability and effectiveness of underwater explosive welding in the combination of Cu and Al, and ensures the performance of Cu/Al composite plate. Yu et al. [26] reported that neither separated nor cracked on the bent specimens including 90 and 180 degrees on the naked eye.

## 3.5  Shear properties – Tensile shearing test

Fig 15 depicts the stress-displacement profiles of Cu/Al composite plate specimens in tensile shearing test. All curves display the same trend of change, indicating that the

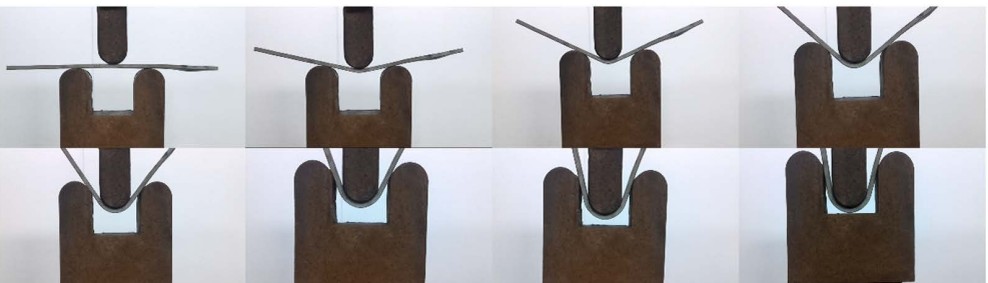

**Fig 13.  Process diagram of bending test for Cu/Al composite plate for underwater explosive welding.**

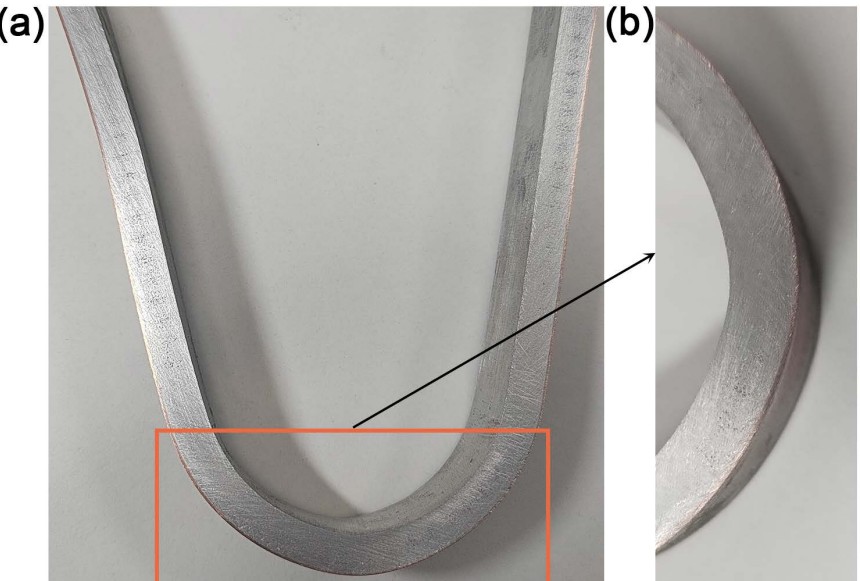

**Fig 14. Morphology of (a) Cu/Al composite plate after bending test and (b) the area with the largest deformation.**

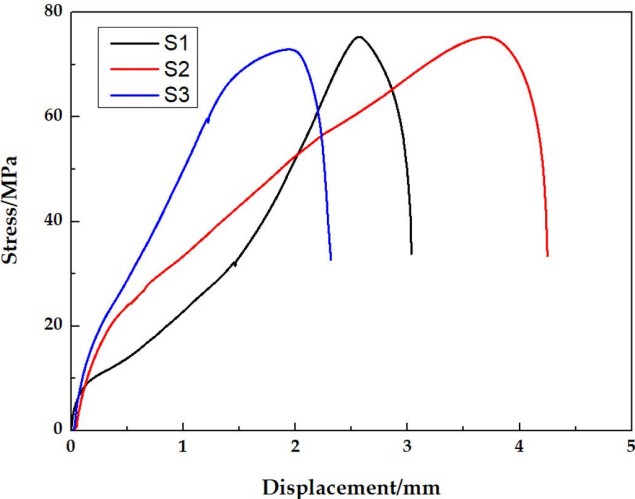

**Fig 15. Stress-displacement curves of Cu/Al composite plate in tensile shearing test.**

bonding between Cu foil and Al plate is stable. The shear strengths of the samples at different positions were 72.9, 75.3 and 75.1 MPa, respectively, indicating excellent bonding strength between Cu foil and Al plate, and the bonding property was consistent in different positions.

Fig 16 shows low and high magnification SEM morphologies of tensile shearing test fracture surfaces on the Cu and Al side of Cu/Al composite plate. There are strips with shear deformation on both sides of Cu and Al. The band spacing and width correspond to the distribution and size of compounds at the Cu/Al bonding interface shown in Fig 8. Fig 17 shows

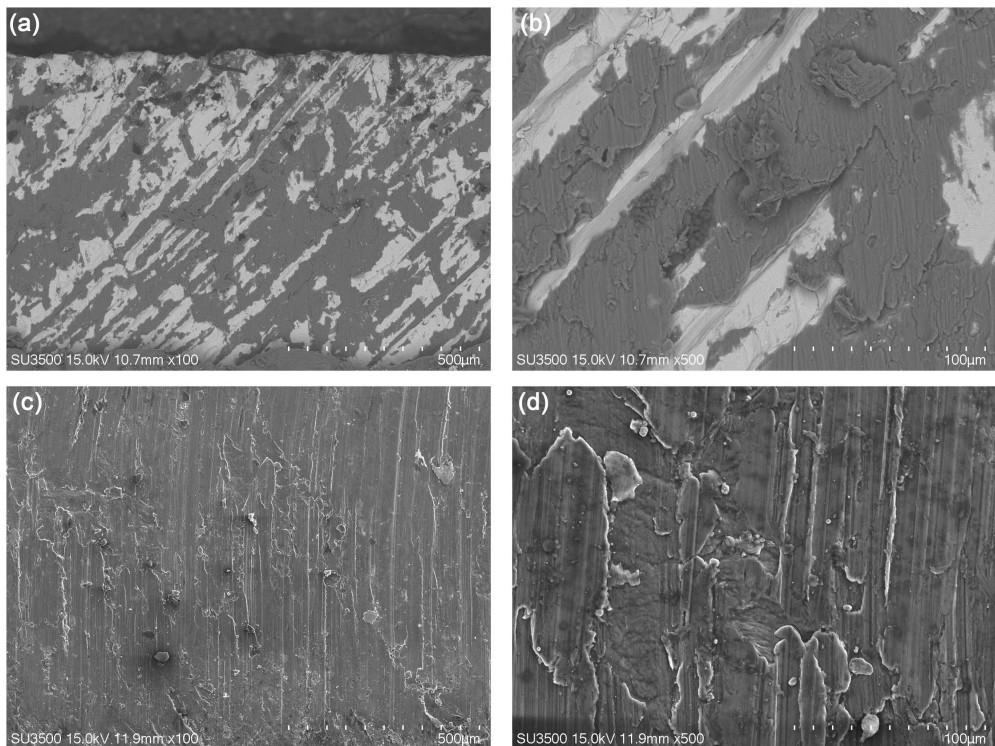

**Fig 16. Low and high magnification SEM morphologies of tensile shearing test fracture surfaces on the Cu (a, b) and Al (c, d) sides.**

the morphology, Al and Cu mapping of tensile shearing test fracture surface on the Cu side. A large amount of Al deformation layer is distributed on the Cu matrix after tensile shearing test. The fracture mainly occurs at the unfused interface or Al matrix near the fusion interface between Cu and Al. According to the tensile shearing teste results of Cu/Al composite plate, the strength of the fusion interface between Cu and Al exceeds that of the Al plate, because the fracture occurs in the low strength material. The strengths of Cu and Al plate are 268 and 98 MPa, respectively, so the fracture at the fusion interface occurs on the Al matrix near the interface.

## 3.6 Microhardness analysis of Cu/Al composite sheet

Fig 18 presents the Vickers microhardness (HV) values measured from fusion interface and different position from it. As shown, hardness of Cu/Al composite plate exhibits a gradient on the Cu and Al side, respectively. The maximum amount of hardness in Cu and Al side occurs at the edge of the fusion. The hardness values of Cu and Al portion reach 135 and 52 HV, respectively. The hardness value close to the interface is higher than other area, this is mainly caused by severe plastic deformation induced by intense shock pressure in these areas [43,44]. The initial hardness values of Cu foil and Al plate employed in the underwater explosive welding are 101 and 42 HV, respectively. Intermetallic compounds (IMCs) were formed, as shown in Fig 8 and marked IMCs region in Fig 18, further contributes to the hardness peak in the fusion interface area, because the hardness value of IMCs at the fusion interface reaches 385 HV.

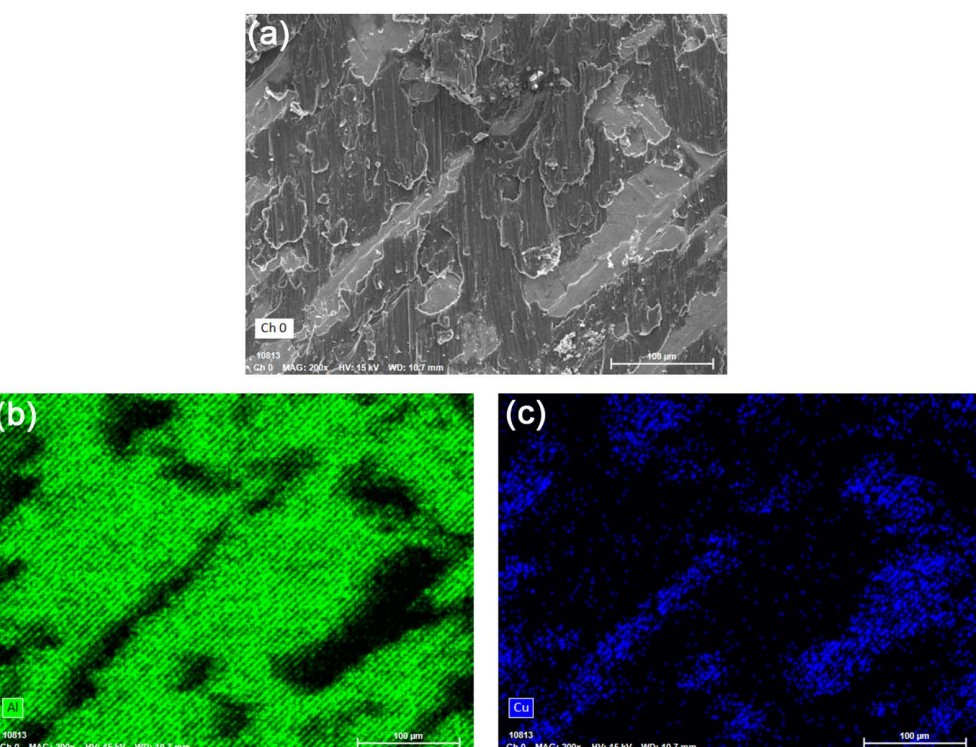

**Fig 17. (a) SEM morphology, (b) Al and (c) Cu element distribution of tensile shearing test fracture surface on the Cu side.**

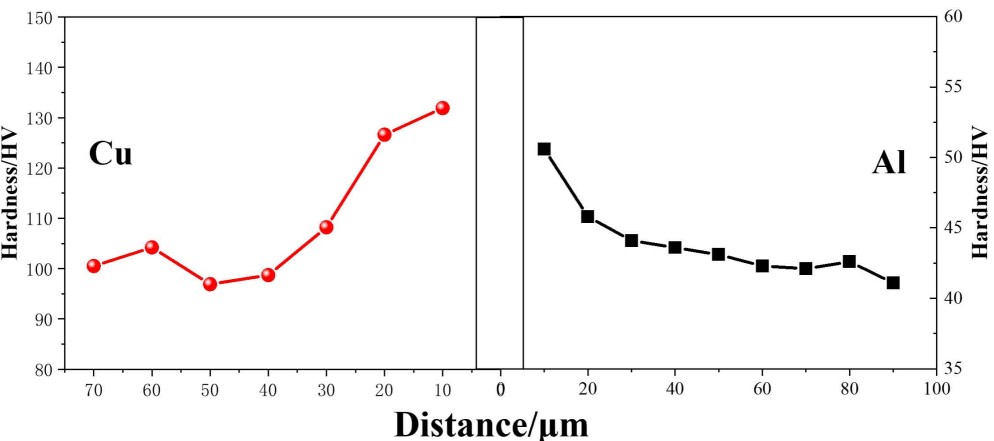

**Fig 18. Micro-hardness profile near the fusion interface of Cu/Al composite plate for underwater explosive welding.**

## 4. Conclusions

Cu/Al composite plate is achieved by underwater explosive welding, at a detonation velocity of 4 000 m/s, when the distance between Cu foil and Al plate is 0.2 mm, and the thickness of Cu foil is only 0.5 mm.

Cu foil and Al plate are bonded well with wavy fusion zones. The tensile strength and tensile shearing strength of Cu/Al composite plate reach 133 and 72.9 MPa, respectively. There is no delamination or cracking at the bonding interface during bending test.

The severed deformation caused by intense shock pressure leads to the hardness of Cu and Al near the interface increase to 135 and 52 HV, respectively. The hardness of the fusion zone even reaches 385 HV.

## Supporting information

**S1 Data.   Original datasets for Fig 11, Fig 15 and Fig 18.**
(ZIP)

## Acknowledgments

We are grateful to Xuzhou Huashun Measurement and Control Company Limited for their help in nondestructive examination.

## Author contributions

**Funding acquisition:** Jinhua Chen.

**Investigation:** Jinhua Chen, Xiaohong Zhou, Hanliang Liang, Peizhong Feng, Yonghao Yu.

**Methodology:** Jinhua Chen, Dapeng Zhou.

**Supervision:** Peizhong Feng.

**Writing – original draft:** Dapeng Zhou, Xiaohong Zhou, Xueqin Kang.

**Writing – review & editing:** Jinhua Chen, Yonghao Yu, Xueqin Kang.

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
