## [Decision Letter · Decision Letter 0]

30 Dec 2024

PONE-D-24-54629Investigation of interface characteristics and mechanical performances of Cu/Al plate fabricated by underwater explosive welding methodPLOS ONE

Dear Dr. Kang,

Thank you for submitting your manuscript to PLOS ONE. After careful consideration, we feel that it has merit but does not fully meet PLOS ONE’s publication criteria as it currently stands. Therefore, we invite you to submit a revised version of the manuscript that addresses the points raised during the review process.

We look forward to receiving your revised manuscript.

Kind regards,

Akash Deep Sharma

Academic Editor

PLOS ONE

Journal Requirements:

[The Key Research and Development Plan of Anhui Province].

4. In the online submission form, you indicated that [The original contributions presented in this study are included in the article; further inquiries can be directed to the corresponding authors].

Reviewers' comments:

Reviewer's Responses to Questions

**Comments to the Author**

1. Is the manuscript technically sound, and do the data support the conclusions?

Reviewer #1: Yes

Reviewer #2: Partly

Reviewer #3: Partly

2. Has the statistical analysis been performed appropriately and rigorously? 

Reviewer #1: Yes

Reviewer #2: Yes

Reviewer #3: N/A

3. Have the authors made all data underlying the findings in their manuscript fully available?

Reviewer #1: Yes

Reviewer #2: Yes

Reviewer #3: Yes

4. Is the manuscript presented in an intelligible fashion and written in standard English?

Reviewer #1: Yes

Reviewer #2: Yes

Reviewer #3: Yes

5. Review Comments to the Author

Reviewer #1: 1.Water bottom is what material should be mentioned.

2.In Fig. 1, the waterproof material is not marked and the material of waterproof is not mentioned.

3.In 3.1, the principle of ultrasonic testing should be introduced, and the red and green color mean?

4.The image ruler is too small in Fig. 5, Fig. 6 and so on, please check all the image ruler of the paper.

5.In Fig 17, please make sure the distance of each measure point, the unit you use is μm.

Reviewer #2: I have reviewed the manuscript title " Investigation of interface characteristics and mechanical performances of Cu/Al plate fabricated by underwater explosive welding method" submitted for possible publication in PLOS ONE. I recommend its publication after some major revisions.

My comments on the manuscript are provided below:

1.It is recommended to mention the research gap in the introduction part.

2.What is the novelty of this study, considering that welding Cu and Al has already been performed using various other welding methods?

3.Discuss what other researchers have done with this combination and explain how your research contributes new and interesting insights.

4.In Section 3.1, specify the materials shown in Figure 5 for better understanding, and provide the length and width of the sample in Figure 5.

5.Is the phased array ultrasonic inspection being performed for the first time on explosive welding samples? If not, include some relevant references.

6.Include an actual image of the welded plates.

7.The scale bars in all the figures are not clearly visible. Additionally, ensure that the names of the materials are mentioned in all the figures.

8.In Figures 15c and 15d, the fracture surface of the shear test sample exhibits some cleavage fracture. However, the observed strength was satisfactory. Considering that pure aluminum was used, should the fracture not exhibit ductile behavior with dimple formation?

9.In microhardness graphs include the hardness of the base metal.

10.A total of 29 references were used, of which 27 are cited in the introduction. Please add more relevant and recent references in the results and discussion section to better support your findings.

11.The conclusions should be generalized and should not repeat content from the abstract or discussion.

Reviewer #3: The article reports the microstructure and mechanical properties of Cu/Al explosive weld joints prepared under water. The article has 17 figures, 2 tables and 29 references. The authors established their claim using OM, SEM, EDS and elemental distribution. The quality of figures are good and the choice of the materials is industrially relevant. Though the authors are appreciated for their attempt, the following critical issues must be addressed before processing further. Few of them are given for authors perusal.

1. It is recommended to include the specific application of the thickness chosen. Further, the methods available for the preparation and how the chosen method is superior than the other methods are to be supplied.

2.Explosive welding of Cu/Al is well reported in literature. How this study adds new knowledge to the scientific community.

3. Novelty of the is a major concern. Why underwater is chosen instead of open air condition

4. It is mandatory to include ASTM standards for mechanical testing performed.

5. Figs.6, 7: Scale bar is not visible. It is recommended to mark salient features on the microstructure.

6. Fig.8 and Table 2: The positioning of elemental distribution attempted is meaningless, as only one data is presented at the interface. It is recommended to add more locations at the interface.

7. Fig.9 does not serve any useful information.

8. More discussion on mechanism of underwater explosive welding is demanded. Placing the setup in a water environment will increase the energy efficiency of the explosive, which will lead to more melting zones at the interface. Is this phenomenon is right. More insights on this critical issue demanded.

9. Value of tensile and shear test specimens are not provided. It is mandatory to provide the mechanical properties of base alloys also.

10. Authors are directed to quote the novel work of this present study.

6. PLOS authors have the option to publish the peer review history of their article (what does this mean?). If published, this will include your full peer review and any attached files.

Reviewer #1: No

Reviewer #2: No

Reviewer #3: No

---

## [Author Response · Author response to Decision Letter 0]

21 Jan 2025

Dear reviewers,

We thank the reviewers for the kind consideration and constructive comments on our manuscript. We have carefully revised the manuscript and provided the point-by-point response below in blue. The changes in the revised manuscript have highlighted in red. We hope these changes will strength our manuscript and fulfill the requirement of PLOS ONE.

1. Thank you for stating the following financial disclosure:

[The Key Research and Development Plan of Anhui Province].

In the Funding section, we have stated the role of the funders. We also revised the content of the Acknowledgements. Hope these changes

can fulfill the requirements of the journal.

The detailed and correct information of the Funding, especially the grant numbers, was provided in the “Funding” Section.

I have not found any information about “Financial Disclosure”. If possible, please revised the “Financial Disclosure” according to the provided “Funding” information. Thank you again for your help.

3. In the online submission form, you indicated that [The original contributions presented in this study are included in the article; further inquiries can be directed to the corresponding authors].

Sorry for not thoroughly understanding the requirements of the journal. I have changed the statement to “All relevant data are within the manuscript and its Supporting Information files”.

4. We note that several of your files are duplicated on your submission. Please remove any unnecessary or old files from your revision, and make sure that only those relevant to the current version of the manuscript are included.

I have removed any unnecessary and old files from our revision.

Reviewer #1:

1.Water bottom is what material should be mentioned.

Thank you for your suggestions. The water bottom was made of sand and worked as anvil and buffer to ensure the stability of the device. This information was added and marked in red in the manuscript.

2.In Fig. 1, the waterproof material is not marked and the material of waterproof is not mentioned.

Thank you for your suggestions. The figure has been revised, with the addition of waterproof and supports. The material of waterproof was polyethylene. This information was added and marked in red in the manuscript.

3.In 3.1, the principle of ultrasonic testing should be introduced, and the red and green color mean?

Thank you for your careful evaluation. The principle of ultrasonic testing was marked in red in 2.3 in the manuscript. The meaning of different color (including red and green) was forgot to export the ruler of the original image. Now, I have exported it and added it to the left side of the image, and the meanings represented by red and green are very intuitive. I also revised the unclear rulers of all images in Figure 5.

4.The image ruler is too small in Fig. 5, Fig. 6 and so on, please check all the image ruler of the paper.

I’m sorry for the inconvenience caused by unclear ruler on the image. I have revised the ruler and checked other images, such as Figure 5, Figure 6, Figure 7 and Figure 9 in the revised manuscript.

5.In Fig 17, please make sure the distance of each measure point, the unit you use is μm.

I’m very sorry. The unit of the horizontal axis is millimeters. I have made modification in the image. I will pay more efforts in the future to avoid such mistakes from happening again. 

Reviewer #2: I have reviewed the manuscript title " Investigation of interface characteristics and mechanical performances of Cu/Al plate fabricated by underwater explosive welding method" submitted for possible publication in PLOS ONE. I recommend its publication after some major revisions.

My comments on the manuscript are provided below:

1.It is recommended to mention the research gap in the introduction part.

Thanks for your suggestion. The research gaps about composite materials fabricated by underwater explosive welding were added and marked in the manuscript.

2.What is the novelty of this study, considering that welding Cu and Al has already been performed using various other welding methods?

Thanks for your suggestions. We have added the existing fabricated methods, such as rolling composite, surface spray, extrusion drawing and vacuum brazing. The problems with these methods also have been listed.

The manuscript explains the merit of underwater explosive welding, because it can achieve the welding of thinner Cu and Al plate. Although there are some researches on underwater explosive welding, most of them focused on the parameters of underwater explosive welding. Few papers have conducted comprehensive research on the bonding interface and properties, especially on the bonding strength of composite materials.

This manuscript fabricated Cu/Al composite materials through underwater explosive welding. Phased array ultrasonic test and tensile shearing test were used for the first time to detect the bonding quality and strength between Cu and Al plate. The manuscript provided a comprehensive characterization of the performance and microstructure of Cu/Al composite materials.

The added and revised contents were marked in red in the manuscript.

3.Discuss what other researchers have done with this combination and explain how your research contributes new and interesting insights.

Other researches on underwater explosive welding only presented the macroscopic or microscopic morphology of composite material cross-sections without studying their bonding interface and properties. Of course, some papers have investigated the performance of composite materials, but only focused on the hardness, tensile, bending or compressive properties. Extensive researches on underwater explosive welding have focused on the distance of flyer plate to lower plate, flyer plate velocity, collision point speed, collision angle and the usage of explosives. Furthermore, the morphology changes and the comprehensive mechanical properties of composited materials should be concentrated, especially the bonding strength between lower and flyer plate.

This manuscript fabricated Cu/Al composite materials through underwater explosive welding. Phased array ultrasonic test and tensile shearing test were used for the first time to detect the bonding quality and strength between Cu and Al plate. The manuscript provided a comprehensive characterization of the performance and microstructure of Cu/Al composite materials.

All the changes were marked in red in the manuscript.

4.In Section 3.1, specify the materials shown in Figure 5 for better understanding, and provide the length and width of the sample in Figure 5.

The width and length of the sample were given and marked in red in the manuscript.

To better determine the location of the test, the actual image of the welded plates was added. We also located the position of the flaw detection.

All the changes were marked in red in the manuscript.

5.Is the phased array ultrasonic inspection being performed for the first time on explosive welding samples? If not, include some relevant references.

Yes. To our knowledge, this is the first time that phased array ultrasonic inspection has been used to detect the bonding quality of composite plate. We referred to some literature on phased array ultrasonic inspection to analyze the results. The referenced papers were added as references and annotated in the manuscript.

6.Include an actual image of the welded plates.

Thanks for your thoughtful reminder. The actual image of the welded plates is very helpful to understand the condition of the Cu/Al composite plate and the location of the test. We have added an actual image of the welded plates as Figure 5 in the manuscript.

7.The scale bars in all the figures are not clearly visible. Additionally, ensure that the names of the materials are mentioned in all the figures.

I’m sorry for the inconvenience caused by unclear scale bars in the images. I have revised the scale bar and checked other images.

I checked all the figures and ensured that the names of the materials are mentioned. All the changes in the image title were marked in red in the manuscript.

8.In Figures 15c and 15d, the fracture surface of the shear test sample exhibits some cleavage fracture. However, the observed strength was satisfactory. Considering that pure aluminum was used, should the fracture not exhibit ductile behavior with dimple formation?

The Cu foil and Al plate slid against each other during the tensile shearing experiment, and the fracture surface of Cu and Al exhibited more plastic deformation characteristics rather than cleavage characteristics.

Meanwhile, the fracture occurred at the intermetallic compounds at the bonding interface, due to the lower strength of Cu/Al composite (~75 MPa) compared to Cu (268 MPa) foil and Al plate (98 MPa). The intermetallic compounds of Cu and Al exhibited more brittle behavior, so the fracture not exhibited ductile behavior with dimple formation.

9.In microhardness graphs include the hardness of the base metal.

The leftmost and rightmost endpoints in Figure 18 represent the hardness of Cu and Al base metals, respectively.

10.A total of 29 references were used, of which 27 are cited in the introduction. Please add more relevant and recent references in the results and discussion section to better support your findings.

Thanks for your reminding. We have added some latest references in the results and discussion section to support our findings. The added references in the References were marked in red and the added contents in the text were also marked in red and annotated with reference.

11.The conclusions should be generalized and should not repeat content from the abstract or discussion.

Thanks for your suggestion. I have rewritten the conclusions. The rewritten conclusion has been generalized according to your suggestion. The rewritten conclusion was marked in red in the manuscript. 

Reviewer #3: The article reports the microstructure and mechanical properties of Cu/Al explosive weld joints prepared under water. The article has 17 figures, 2 tables and 29 references. The authors established their claim using OM, SEM, EDS and elemental distribution. The quality of figures are good and the choice of the materials is industrially relevant. Though the authors are appreciated for their attempt, the following critical issues must be addressed before processing further. Few of them are given for authors perusal.

1. It is recommended to include the specific application of the thickness chosen. Further, the methods available for the preparation and how the chosen method is superior than the other methods are to be supplied.

Cu is a strategic resource due to its low content (only 0.01 wt.% of the Earth’s crust) and difficulty in extracting and refining. The thinner of the Cu in composite materials, the more Cu material is saved. Therefore, it is necessary to explore methods for thinner Cu composites. We have added these contents in the manuscript and marked in red.

We also added other methods for preparing Cu/Al composite materials and listed their shortcomings.

The manuscript explains the merit of underwater explosive welding, because it can achieve the welding of thinner Cu and Al plate.

All the changes were marked in red in the manuscript.

2.Explosive welding of Cu/Al is well reported in literature. How this study adds new knowledge to the scientific community.

Yes. Explosive welding of Cu/Al is well reported in literature. The explosive welding of this study was carried our underwater. Although there are some researches on underwater explosive welding, most of them focused on the parameters of underwater explosive welding. Few papers have conducted comprehensive research on the bonding interface and properties, especially on the bonding strength of composite materials.

This manuscript fabricated Cu/Al composite materials through underwater explosive welding. Phased array ultrasonic test and tensile shearing test were used for the first time to detect the bonding quality and strength between Cu and Al plate. The manuscript provided a comprehensive characterization of the performance and microstructure of Cu/Al composite materials.

The added and revised contents were marked in red in the manuscript.

3. Novelty of the is a major concern. Why underwater is chosen instead of open air condition

The manuscript explains the merit of underwater explosive welding, because water acts as a transport medium to generate a uniform pressure. The incompressibility of water is much higher than that of air. These characteristics of water ensure that the explosive force does not generate high temperature in water, and it does enable the water to move forward together with the foil metals, and maintaining their integrity. Underwater explosive welding can achieve the welding of thinner Cu and Al plate compared with open air condition.

The explanations were added and marked in red in the manuscript.

4. It is mandatory to include ASTM standards for mechanical testing performed.

Thanks for your suggestion. The tensile, bending and hardness tests of Cu, Al and Cu/Al were conducted according to the standards ASTM E8/E8M-22, ASTM E290-22 and ASTM E92-23. I have added this information in the manuscript and marked in red.

There are no relevant standards to test the bonding strength of Cu foil and Al plate due to the thickness of Cu foil. The tensile shearing test was adopted to reflect the bonding performance of Cu foil and Al plate. The manuscript provides a detailed explanation of the sample and testing procedure.

5. Figs.6, 7: Scale bar is not visible. It is recommended to mark salient features on the microstructure.

I’m sorry for the inconvenience caused by unclear ruler on the image. I have revised the ruler and checked other images, such as Figure 5, Figure 6, Figure 7 and Figure 9 in the revised manuscript.

6. Fig.8 and Table 2: The positioning of elemental distribution attempted is meaningless, as only one data is presented at the interface. It is recommended to add more locations at the interface.

The different color in SEM image (Figure 9a in the revised manuscript) and the element distribution in EDS images (Figure 9b, 9s and 9d in the revised manuscript) indicate diffusion between Cu and Al. The point analysis shows the formation of compounds at the interface between Cu and Al.

More locations were selected in our future study to deduce the evolution of the Cu/Al cross-section during the subsequent heat treatment.

Thanks for your suggestion. It not only improves the quality of this manuscript, but also points out the main content of our future research.

7. Fig.9 does not serve any useful information.

I’m sorry for the inconvenience caused by unclear metallographic structure in the image. The metallographic corrosion of Cu/Al composite is difficult to achieve due to the huge difference in corrosio

---

## [Decision Letter · Decision Letter 1]

4 Feb 2025

PONE-D-24-54629R1Investigation of interface characteristics and mechanical performances of Cu/Al plate fabricated by underwater explosive welding methodPLOS ONE

Dear Dr. Kang,

Thank you for submitting your manuscript to PLOS ONE. Your revised manuscript with required reviews has now been completed. You can see in your author dashboard that two of the reviewers are completely satisfied with your responses whereas one of the reviewer is still suggesting the academic meritorious points that can be added to your manuscript for better understanding to the readers. Please note our reviewers play a very vital role despite of their over-occupancies for evaluating the manuscripts. So, being an editor I deem it wise to respect every single comments by the respect-worthy reviewers. So, you are requested to revise your manuscript and revise the current manuscript.

We look forward to receiving your revised manuscript.

Kind regards,

Akash Deep Sharma

Academic Editor

PLOS ONE

Journal Requirements:

Reviewers' comments:

Reviewer's Responses to Questions

**Comments to the Author**

1. If the authors have adequately addressed your comments raised in a previous round of review and you feel that this manuscript is now acceptable for publication, you may indicate that here to bypass the “Comments to the Author” section, enter your conflict of interest statement in the “Confidential to Editor” section, and submit your "Accept" recommendation.

Reviewer #1: (No Response)

Reviewer #2: (No Response)

Reviewer #3: All comments have been addressed

2. Is the manuscript technically sound, and do the data support the conclusions?

Reviewer #1: (No Response)

Reviewer #2: Yes

Reviewer #3: Yes

3. Has the statistical analysis been performed appropriately and rigorously? 

Reviewer #1: (No Response)

Reviewer #2: Yes

Reviewer #3: N/A

4. Have the authors made all data underlying the findings in their manuscript fully available?

Reviewer #1: (No Response)

Reviewer #2: Yes

Reviewer #3: Yes

5. Is the manuscript presented in an intelligible fashion and written in standard English?

Reviewer #1: (No Response)

Reviewer #2: Yes

Reviewer #3: Yes

6. Review Comments to the Author

Reviewer #1: (No Response)

Reviewer #2: The authors have adequately addressed the reviewer’s comments, and the manuscript is recommended for acceptance after minor revisions.

Earlier Comment No. 5: Is the phased array ultrasonic inspection being performed for the first time on explosive welding samples? If not, include some relevant references.

The authors have mentioned: Yes, to our knowledge, this is the first time that phased array ultrasonic inspection has been used to detect the bonding quality of composite plate.

Thank you for your clarification. I understand that you may not have found prior studies on this topic. However, one study has utilized phased array ultrasonic inspection in explosive welding. Please review it, as it may be useful for your literature.

1. Upadhyay, Abhishek, et al. "Phased array-based ultrasonic testing of explosively welded aluminium and stainless steel plates." Transactions of the Indian Institute of Metals 72.6 (2019): 1521-1525. https://doi.org/10.1007/s12666-019-01603-5

Reviewer #3: The authors have responded to the queries raised in a positive manner. In addition, they made efforts to incorporate the same in the manuscript.

7. PLOS authors have the option to publish the peer review history of their article (what does this mean?). If published, this will include your full peer review and any attached files.

Reviewer #1: No

Reviewer #2: No

Reviewer #3: No

---

## [Author Response · Author response to Decision Letter 1]

14 Feb 2025

Dear reviewers,

We thank the reviewers’ recognition of our work and valuable recommendation. We have carefully revised the manuscript and provided the point-by-point response below in blue. The changes in the revised manuscript have highlighted in red. We hope these changes will strength our manuscript and fulfill the requirement of PLOS ONE.

Reviewer #1: (No Response)

Reviewer #2: The authors have adequately addressed the reviewer’s comments, and the manuscript is recommended for acceptance after minor revisions.

Earlier Comment No. 5: Is the phased array ultrasonic inspection being performed for the first time on explosive welding samples? If not, include some relevant references.

The authors have mentioned: Yes, to our knowledge, this is the first time that phased array ultrasonic inspection has been used to detect the bonding quality of composite plate.

Thank you for your clarification. I understand that you may not have found prior studies on this topic. However, one study has utilized phased array ultrasonic inspection in explosive welding. Please review it, as it may be useful for your literature.

1. Upadhyay, Abhishek, et al. "Phased array-based ultrasonic testing of explosively welded aluminium and stainless steel plates." Transactions of the Indian Institute of Metals 72.6 (2019): 1521-1525. https://doi.org/10.1007/s12666-019-01603-5

Sorry for not finding this valuable paper. This reference if very helpful for analyzing the bonding performance of Cu/Al composite plate. I cited this paper as Ref. 35 in our manuscript and changed the numbering of other references. Thank you again for your suggestion.

Reviewer #3: The authors have responded to the queries raised in a positive manner. In addition, they made efforts to incorporate the same in the manuscript.

Thank you for acknowledging our work. You earlier opinions have made our work more comprehensive, and have great help to our research.

Thank you again for your recognition of our work and valuable recommendation to improve the quality of our manuscript.

Kind regards,

Xueqin Kang

---

## [Decision Letter · Decision Letter 2]

27 Feb 2025

Investigation of interface characteristics and mechanical performances of Cu/Al plate fabricated by underwater explosive welding method

PONE-D-24-54629R2

Dear Dr. Kang

We’re pleased to inform you that your manuscript has been judged scientifically suitable for publication and will be formally accepted for publication once it meets all outstanding technical requirements.

Kind regards,

Akash Deep Sharma

Academic Editor

PLOS ONE

Additional Editor Comments (optional):

Reviewers' comments:

Reviewer's Responses to Questions

**Comments to the Author**

1. If the authors have adequately addressed your comments raised in a previous round of review and you feel that this manuscript is now acceptable for publication, you may indicate that here to bypass the “Comments to the Author” section, enter your conflict of interest statement in the “Confidential to Editor” section, and submit your "Accept" recommendation.

Reviewer #2: All comments have been addressed

2. Is the manuscript technically sound, and do the data support the conclusions?

Reviewer #2: (No Response)

3. Has the statistical analysis been performed appropriately and rigorously? 

Reviewer #2: (No Response)

4. Have the authors made all data underlying the findings in their manuscript fully available?

Reviewer #2: (No Response)

5. Is the manuscript presented in an intelligible fashion and written in standard English?

Reviewer #2: (No Response)

6. Review Comments to the Author

Reviewer #2: The author has successfully addressed all the comments raised by the reviewer. The necessary revisions have been made, and the paper can now be accepted for publication.

7. PLOS authors have the option to publish the peer review history of their article (what does this mean?). If published, this will include your full peer review and any attached files.

Reviewer #2: No

---

## [Editor Report · Acceptance letter]

PONE-D-24-54629R2

PLOS ONE

Dear Dr. Kang,

I'm pleased to inform you that your manuscript has been deemed suitable for publication in PLOS ONE. Congratulations! Your manuscript is now being handed over to our production team.

Kind regards,

on behalf of

Dr. Akash Deep Sharma

Academic Editor

PLOS ONE